# African Swine Fever Shock: China’s Hog Industry’s Resilience and Its Influencing Factors

**DOI:** 10.3390/ani13182817

**Published:** 2023-09-05

**Authors:** Zizhong Shi, Xiangdong Hu

**Affiliations:** Institute of Agricultural Economics and Development, Chinese Academy of Agricultural Sciences, Beijing 100081, China; shizizhong@caas.cn

**Keywords:** African swine fever, hog epidemic, animal disease, hog industry, industry resilience, resistance, recoverability

## Abstract

**Simple Summary:**

The occurrence of African swine fever has restricted the sustainable and healthy development of the hog industry and its ability to enhance the supply of pork, thus negatively impacting China’s economic and social development. During 2018–2021, the resilience of China’s hog industry improved due to the adjustments to the prevention and control mechanisms in African swine fever policy and the subsequent recovery of hog breeding and pork production. There are differences among the key factors influencing the resilience of the hog industry across different provinces and periods. These factors include the slaughter rate, carcass weight, mortality rate, culling rate, economic level, industrial structure, per capita consumption, scale level, resource carrying, etc. The aim of this study is to provide guidance for better prevention and mitigation of epidemic risks, thus promoting the high-quality development of the hog industry and meeting the nutritional needs of residents.

**Abstract:**

African swine fever has damaged the foundation of China’s hog industry, caused a serious decline in hog production, highlighted the contradiction between supply and demand in the pork market, and led to major economic and social impacts. The industrial resilience of 31 Chinese provinces to African swine fever shock and its spatial and temporal differentiation characteristics from 2018 to 2021 were measured in this study from the two dimensions of resistance and recoverability. Using Geodetector, the key factors influencing the resilience of China’s hog industry were explored. The results showed that 2018–2019 and 2020–2021 represented the resistance and recovery periods of the hog industry under African swine fever shock, respectively, with poor resilience characterizing the resistance period and improved resilience characterizing the recovery period. At the early stages of the African swine fever outbreak, the hog industries in Tianjin, Shanxi, Guangxi, and Yunnan had robust resistance due to the slaughter rate, economic level, mortality rate, carcass weight, and culling rate in those areas. At the most severe stage of the outbreak, resistance was generally poor in all provinces due to the slaughter rate, per capita consumption, and scale level at the time. During the period of rapid recovery in hog production, the recoverability of each province was very strong due to the industrial structure, culling rate, economic level, and resource carrying capacity at that time. During the reasonable adjustment period of hog production capacity, the recoverability based on the breeding sow inventory in 13 provinces, including Henan, Shandong, and other large hog-breeding provinces, was negative due to the scale level, slaughter rate, per capita consumption, and resource carrying at that time. Taking measures to enhance the resilience of the hog industry, strengthen the prevention and control of hog epidemics, improve the monitoring and early warning mechanisms, and enhance the ability of the hog industry to cope with major animal epidemics is recommended.

## 1. Introduction

According to a traditional Chinese saying, “hogs and grain can stabilize the nation”. The hog industry occupies a key strategic position in China’s national economic and social development. Since the reform and opening up, the development of China’s hog industry has achieved remarkable results, but it still faces many outstanding problems and practical challenges. In particular, the outbreak of African swine fever in 2018 had a significant negative impact on both the hog industry and the pork market, severely challenging the foundation of the industry, leading to a severe decline in hog production, exacerbating the contradiction between supply and demand in the pork market, and pushing up the prices of livestock products across the board. In 2019, the number of hogs slaughtered began to fall, declining by 21.6% year on year, and pork production fell by 21.3%. In addition to the impact of cyclical factors, the effect of the African swine fever shock on hog price fluctuations that began in the second half of 2019 was exceptionally obvious, with prices surging and becoming an important factor driving up China’s consumer price index (CPI). At the same time, African swine fever caused major problems, such as a shortage of breeding sows, the retention of ternary sows, and the low productivity of sows in general, which severely impaired the sustainable development of the hog industry. Compared to the hog epidemic and avian influenza of previous years, the impact of African swine fever was much greater and deeper and had more obvious and negative effects on China’s national economic development, as well as on the production and lives of the Chinese people.

To cope with the negative impact of African swine fever and accelerate the recovery of hog production, the Chinese government has intensively launched a series of policy initiatives, including infrastructure, agricultural machinery purchase, manure resource utilization, standardization demonstration, resumption of production support, epidemic prevention and control, finance and insurance, etc., which have entailed breakthroughs in terms of land use, prohibition, and breeding restriction policies. In 2022, China’s hog inventory reached 452.56 million heads, which was up 0.7% from the previous year, when the breeding sow inventory reached 43.90 million heads, representing an equivalent increase of 1.4%; hog slaughter reached 699.95 million heads, increasing by 4.3%; pork production reached 55.41 million tons, increasing by 4.6%; and hog production capacity was set within a reasonable range. According to another traditional Chinese saying, “Family money with animals does not count”, referring to the major risk of animal disease incurred in livestock breeding. At present, China still faces challenges, including a wide range of diseases, considerable capacity issues with respect to total hog breeding capacity, weak epidemic prevention and control, and a large risk of the introduction of foreign animal epidemics into the country. The future prevention and control of major animal epidemics cannot be ignored. In this context, to accelerate the improvement of hog industry resilience, building a safe and effective pork market “defense system” is of great practical importance.

There are currently no domestic or foreign studies on the resilience of China’s hog industry and its influencing factors, and the related literature has placed a greater focus on exploring the relevant impact mechanisms. Studies have concluded that epidemics exert a high negative impact, can cause severe economic losses, and pose a major threat to the livelihoods of farming households in developing countries [1,2,3]. Epidemics can affect market movements through both supply and demand levels, including constraining production, reducing consumption, and regulating trade [4,5,6,7,8]. African swine fever has reduced hog production capacity, tightened supply, and initiated a new cycle of price increases that, in turn, has contributed to the increase in the CPI [9,10]. The impact of epidemics such as African swine fever also manifests in micro aspects, such as the low willingness of farmers to expand the scale of their farming under the epidemic shock, with less than one-quarter of such farmers seeking to expand the scale of their farming [11]. The degree of epidemic loss has a significant negative impact on the decision-making behavior of farmers, but the difference in the factors that affect the decision-making behavior across different farm sizes is obvious [12]. This difference also affects the behavior, efficiency, and effectiveness of the farm’s disease prevention and control efforts [13,14].

Epidemics have a major impact on price fluctuations in the hog market, and “behind every round of large swings in hog prices, there is a lingering shadow of epidemic disease” [15,16,17]. Uncertain events, such as epidemics, are an important cause of large fluctuations in hog prices and their cyclical, time-varying, and state transition characteristics [18,19,20,21]. This type of unexpected event information has a leverage effect on hog price fluctuations, which is why hog price fluctuations exhibit the characteristics of clustering, asymmetry, memory, and persistence [22]. The impact of this type of shock extends to the entire industrial chain and related markets, albeit with significant regional differences [23,24]. Epidemic shocks not only affect supply, demand, and price changes in the hog market but can also lead to a lack of confidence in the administration of the market by market participants [25]. China’s hog industry still faces a relatively large epidemic risk, the epidemic prevention and control policy system is not sound, the prevention and control capacity is not strong, and urgent attention is needed [26,27].

Thus far, the literature on resilience has focused on the agricultural and macroeconomic fields, and its research paradigm and research methods have important reference value for this study. From the perspective of agricultural resilience, factor analysis, coupling coordination degree, and other methods have been used in the literature to study the characteristics, laws, and synergistic strength of agricultural economic resilience and high-quality agricultural development [28]; the entropy value-TOPSIS model and the superefficient SBM model were adopted to measure the resilience and efficiency of China’s marine fishery economy [29]; and an evaluation was constructed based on the three dimensions of resistance, adaptability, and changeability, as measured by the index system, thus exploring the effect and mechanism of the digital economy on the robustness of the agricultural economy [30]. There are also studies that have focused on the theoretical and practical logic of China’s agricultural resilience, exploring the spatial and temporal characteristics of agricultural resilience and its influencing factors [31,32,33,34]. In terms of nonagricultural resilience, studies have been conducted to measure the resilience of Beijing’s tourism industry and China’s foreign trade from the perspectives of resistance, recoverability, and resilience enhancement capacity by using the core variable method against the background of the impact of COVID-19 [35,36], to measure the resilience of China’s provincial economy and the key factors affecting it from the perspective of resistance [37], and to explore the spatial and temporal heterogeneity of logistics timeliness from a resilience perspective [38].

Considering that the resilience of China’s hog industry has not been systematically measured in the literature, and that the relevant influencing factors have yet to be explored, examining resilience is of great practical importance for promoting the stable and orderly development of the hog industry and ensuring the safe and effective supply of the pork market. Therefore, two core variables, namely, the inventories of hogs and breeding sows, were selected for this study, and the industrial resilience and spatial and temporal characteristics of 31 provinces that faced the impact of African swine fever in 2018–2021 were systematically measured. In the study, the two dimensions of resistance and recoverability are used, the influencing mechanism of the resilience of China’s hog industry is investigated through Geodetector, and relevant countermeasures and recommendations for policy and production decisions are finally proposed.

## 2. Methods and Materials

### 2.1. Methods

#### 2.1.1. Hog Industry Resilience Measurement Method

Based on the theory of regional economic resilience, two indicators were selected—namely, the inventory of hogs, and that of breeding sows—as the core variables of this study, used to measure the resilience of the hog industry to the impact of African swine fever by measuring its resistance and recoverability. Resistance focuses on the ability of the hog industry to avoid the decline caused by the impact of African swine fever, while recoverability focuses on the ability of the hog industry to adapt and recover through its own vitality, external thrust, and other factors. Under the impact of African swine fever, this study focuses on the resistance of the hog industry to falling from its normal level and the recoverability of the hog industry from such a fall back to its normal level.

The formula for calculating resistance is as follows:(1)$$Resistance_{r}=\frac{V_{i}^{res}-V_{i-1}^{res}}{V_{i-1}^{res}}$$
where *Resistance_r_* denotes the resistance of the hog industry, *V_i_^res^* denotes the inventory of hogs or breeding sows in province *r* in year *i* during the resistance period, and *V_i−_*_1_*^res^* denotes the inventory of hogs or breeding sows in province *r* in year *i* − 1 during the resistance period.

The formula for calculating recoverability is as follows:(2)$$Recoverability_{r}=\frac{V_{i}^{rec}-V_{i-1}^{rec}}{V_{i-1}^{rec}}$$
where *Recoverability_r_* denotes the recoverability of the hog industry, *V_i_^rec^* denotes the inventory of hogs or breeding sows in province *r* in year *i* during the recovery period, and *V_i−_*_1_*^rec^* denotes the inventory of hogs or breeding sows in province *r* in year *i* − 1 during the recovery period.

#### 2.1.2. Geodetector

In this study, Geodetector was used to detect the relationships between various factors and the resilience of the hog industry in each province of China, and to search for the factors with the most explanatory power. Geodetector is a statistical method used to detect spatial heterogeneity and reveal the driving force(s) behind it, including factor detection, interaction detection, risk area detection, and ecological detection [39]. The specific model of Geodetector is as follows:(3)$$q=1-\frac{1}{N\sigma^{2}}\sum_{h=1}^{L}N_{h}\sigma_{h}^{2}$$
where *q* denotes the driving force explanatory factor, which has a value range of 0–1. A larger value indicates that the influencing factor has a higher degree of explanation regarding the resilience of the regional hog industry; *h* = 1,2 …; *L* is the number of classifications; *N_h_* and *N* denote the number of sample units in stratum *h* and in the whole region, respectively; and *σ_h_*^2^ and *σ*^2^ denote the variance in stratum *h* and in the whole region, respectively.

To explore the key factors that affect the resilience of the hog industry, regional economic theory, the actual development of the hog industry, and the existing research foundation were combined for this study, and the four dimensions of the development foundation, scientific and technological support, basic security, and epidemic shock were analyzed. Regarding the development foundation, four main indicators were considered: economic level, industrial structure, market share, and per capita pork consumption. Regarding scientific and technological support, four main indicators were considered: hog slaughter rate, carcass weight, scale breeding degree, and labor productivity. Regarding basic security, three main indicators were considered: the comparative efficiency of hog breeding, resource carrying capacity, and technical service level. Finally, regarding the epidemic shock, three main indicators were considered: the number of cases, mortality, and the culling rate of African swine fever. In general, the better the development foundation of the hog industry, the higher the level of scientific and technological support and basic security, and the more robust the resilience of the hog industry. Furthermore, the more intense the impact of African swine fever, the weaker the resilience of the hog industry. The relevant influencing factors and the methods for calculating the resilience of China’s hog industry are shown in Table 1.

### 2.2. Materials

The period of 2018–2021 was chosen as the research period because the African swine fever outbreak began in 2018, while by 2021 the epidemic was better controlled and hog production had recovered to its previous normal levels. The study unit consisted of 31 provinces in China, excluding Hong Kong, Macau, and Taiwan. The basic data for this study were obtained from the *China Statistical Yearbook*, the *China Animal Husbandry and Veterinary Yearbook*, the National Compilation of Cost and Benefit Information of Agricultural Products, and the *Official Veterinary Bulletin of the Ministry of Agriculture and Rural Affairs of the People’s Republic of China*. Some of the data were specifically treated in the following ways: (1) Tibet lacks market prices for hogs and corn, so they were replaced in this study with the corresponding average prices across the country. (2) Labor productivity in hog breeding is measured by the ratio of total output value per head to the average number of laborers on large, medium, and small farms; however, there is a lack of data for the small farms in Beijing, Shanghai, and Tibet. For medium farms, there is a lack of data for Shanghai and Tibet. Finally, for large farms, there are data gaps for Tibet and Ningxia. Therefore, national average data were utilized to fill these gaps in this study.

## 3. Results and Discussion

### 3.1. Spatial and Temporal Characteristics of Hog Industry Resilience

Before examining the resilience of the hog industry and its influencing factors, it was necessary to identify the resistance and recovery periods for the hog industry. After the African swine fever outbreak was first diagnosed in China’s Liaoning Province in August 2018, it spread rapidly across the country in just eight months. The inventories of hogs and breeding sows decreased by 3.04% and 4.71% in 2018 compared to the previous year, respectively, and by 27.50% and 27.70% in 2019 compared to the previous year, respectively. At the end of 2019, the Chinese government formulated and implemented the Three-Year Action Program for Accelerating the Resumption of the Development of Hog Production to promote the accelerated recovery of hog production to its normal levels in previous years. In 2020, the inventories of hogs and breeding sows increased by 30.96% and 35.09%, respectively, compared to the previous year, and the production capacity of hogs effectively recovered to a normal level in 2021, increasing by 10.51% and 4.02%, respectively, compared to the previous year (Figure 1). Therefore, 2018–2019 can be defined as the resistance period of the hog industry under the impact of African swine fever, during which the resilience of the hog industry was obviously poor; 2020–2021 can be dubbed the recovery period of the hog industry under the impact of African swine fever, during which the resilience of the hog industry became more robust.

Based on this, and considering the hog industry resilience measurement method and basic data, the resilience of the hog industry and its spatial and temporal characteristics were calculated. Table 2 shows the spatiotemporal characteristics of the resistance and recoverability of the hog industry, while Figure 2 and Figure 3 show the spatiotemporal characteristics of the industry’s resilience based on the inventories of hogs and breeding sows, respectively.

In terms of resistance, in the early stages of the African swine fever outbreak, i.e., 2018, Tianjin, Shanxi, Guangxi, and Yunnan all had positive resistance, and the positive resistance of Shanxi and Xinjiang was based on the inventory of breeding sows, indicating that the hog industry in these provinces was not negatively affected by African swine fever and maintained a relatively normal development trend. The five provinces with the worst resistance were Beijing, Shanghai, Fujian, Ningxia, and Tibet, which are all areas with relatively weak hog breeding bases; Liaoning Province, where African swine fever was first diagnosed, was also hit relatively hard. The main reason for this was the outbreak of African swine fever that occurred in the second half of 2018, which spread quickly and widely in a relatively short period. During that period, however, the epidemic had not yet spread to all parts of the country, and there were still some areas that had not been affected or were less affected by the epidemic.

In 2019, when African swine fever was at its most severe state, aside from the positive resistance level of the hog industry in Ningxia based on the inventory of breeding sows, the resistance of other provinces as based on hog or breeding sow inventories was negative, indicating that the resistance of the hog industry was generally poor during the most severe period of the epidemic. The main reason for this was that African swine fever had appeared in China for the first time, and neither government departments nor hog farmers had mature knowledge or experience in epidemic prevention and control. This fact, coupled with irrational epidemic prevention and control policies and a poor level of epidemic awareness, caused a more serious market panic and artificially deepened the impact of African swine fever. The five provinces with the worst resistance based on hog inventories were Beijing, Jiangsu, Hainan, Qinghai, and Shanghai, and the five provinces with the worst resistance based on breeding sow inventories were Beijing, Hainan, Qinghai, Jiangsu, and Guangdong. From the point of view of large hog-farming provinces, Hunan, Sichuan, and Yunnan Provinces were affected by the impact of African swine fever to varying degrees, but the resistance was high, and the impact was contained to a relatively moderate level.

In terms of recoverability, when the recovery of hog production accelerated, in 2020, the recoverability of the hog industry in 31 provinces was positive based on the inventories of hogs or breeding sows, indicating that the hog industry nationwide had been showing continuous accelerated recovery and had strong recoverability. The top five provinces in terms of recoverability based on their hog inventories were Beijing, Jiangsu, Qinghai, Shanghai, and Tibet, and the provinces at the bottom of this ranking were Shaanxi, Jilin, Guangxi, Guizhou, and Heilongjiang; the top five provinces in terms of recoverability based on the breeding sow inventory were Qinghai, Jiangsu, Beijing, Shanghai, and Hainan, and the provinces at the bottom of this ranking were Tibet, Shaanxi, Guangxi, Jilin, and Yunnan. The reason for the generally strong resilience of the hog industry in 2020 was mainly that the government had implemented a series of policies to prevent and control African swine fever and had vigorously promoted and supported hog farming, which included the issuance and implementation of policy documents such as the Three-Year Action Program for Accelerating the Resumption of the Development of Hog Production, which resulted in a very high level of activism among local governments and large-scale farms, aimed at resuming and expanding hog farming. During the piglet shortage period, for the rearing and expansion of the purchase of hogs, ternary sows or backcross sows bred as binary sows, crossbred sows bred as purebred hogs, and other phenomena became more frequent among many farm households; moreover, the birth of foreign piglets was pursued by domestic hog farms.

In 2021, two provinces showed negative recoverability based on their inventory of hogs, namely, Ningxia and Shanghai, while the recoverability based on the inventory of breeding sows was relatively more negative, spanning 13 provinces, including Henan, Shandong, and other large hog-breeding provinces. The main reason for this is that, in 2020, the year of accelerated recovery of hog production, many farm households blindly and disorderedly expanded the healthy development of the hog industry, which caused potentially severe problems. It has also been factually demonstrated that due to the unregulated expansion of hog farming, hog market prices in 2021 appeared to fall off a cliff and experienced the strongest “hard landing”, which resulted in many farms experiencing the “epidemic impact loss–high price profits–price plunge loss” “roller coaster” of the “second blow”. To ensure the orderly development of the hog industry, the Ministry of Agriculture and Rural Affairs of the People’s Republic of China issued and implemented the Hog Production Capacity Control Implementation Plan (Provisional) in 2021, which clearly proposed stabilizing the hog production capacity within a reasonable range. The 14th Five-Year Plan period had the effect of stabilizing the normal retention of breeding sows at approximately 41 million, with a minimum retention of no less than 37 million, causing many provinces to reduce the numbers of breeding sows.

### 3.2. Factors Influencing Hog Industry Resilience

In this study, the resistance and recoverability of the hog industry in each province was taken as the dependent variable, and 14 indicators in four dimensions, i.e., development foundation, scientific and technological support, basic security, and epidemic shock, were taken as independent variables. After the data of the independent variables were discretized by using the natural breakpoint method of ArcGIS, we applied the factor detection and interaction detection of Geodetector to analyze the resistance and recoverability of the hog industry in China. The spatial differentiation of the factors affecting the resistance and recoverability of China’s hog industry was also analyzed. Table 3 shows the results of the factor detection of resistance and recoverability of the hog industry, and Table 4, Table 5, Table 6, Table 7, Table 8, Table 9, Table 10 and Table 11 show the results of the interaction detection of the resistance and recoverability of the hog industry.

Regarding the factors that influence resistance in the hog industry, in the early stages of the African swine fever outbreak in 2018, the five indicators with the highest explanatory power regarding resistance based on the inventory of hogs were the slaughter rate, economic level, mortality rate, carcass weight, and culling rate, while the five indicators with the highest explanatory power regarding resistance based on the inventory of breeding sows were the slaughter rate, economic level, carcass weight, mortality rate, and culling rate. Contrasting these two sets of factors, one can see that they appear to be the same indicators, revealing that the scientific and technological support capacity, the economic development level, and the epidemic shock had a large impact on the resistance of the hog industry in the early stages of the African swine fever outbreak. The higher the level of economic development, the more fundamental advantages the development of the hog industry has, and the stronger its resistance. The more severe the African swine fever shock, the greater the impact on the hog industry and the weaker its resistance. In general, the slaughter rate and carcass weight reflect the level of scientific and technological development, and these factors show a positive relationship with the resilience of the hog industry, i.e., the higher the slaughter rate and the carcass weight, the stronger the resilience of the hog industry. However, the extent of the impacts in the early stages of the African swine fever outbreak cannot be solely based on the general situation, as this extent is affected by information asymmetry and many other factors. Many farmers who were experiencing market panic and other influences accelerated their slaughter activities, which resulted in a substantial increase in the hog slaughter rate and a sharp decrease in carcass weight. The slaughter rate and the carcass weight at that time both demonstrated that simply increasing the level of industry resilience is not sufficient.

In 2019, when the African swine fever outbreak was at its worst, the five indicators with the highest explanatory power for resistance based on the hog inventory were the slaughter rate, per capita consumption, scale level, economic level, and the number of cases, while the five indicators with the highest explanatory power for resistance based on the breeding sow inventory were the per capita consumption, slaughter rate, technical service, labor productivity, and scale level. When contrasting these two sets of indicators, it can be seen that the indicators for both resistance bases were the slaughter rate, per capita consumption, and scale level. As mentioned above, when an outbreak of African swine fever occurs, many hog farmers accelerate their slaughter efforts, resulting in a different slaughter rate from that of normal years, and the abnormal slaughter level of hog farming is directly related to a lack of resilience. During the period of African swine fever’s most serious impact, large-scale farms had a better foundation and stronger advantages than small-scale farms and free-range farmers, and their resistance to the impact of African swine fever was greater. This is because the higher the scale level, the greater the resistance and resilience of the hog industry. The epidemic impact also caused a decline in hog production capacity that led to pork supply shortages, and residents’ consumption and other demand-side factors became important factors affecting the resistance of the hog industry.

Regarding the factors influencing the recoverability of the hog industry, in 2020, which was the year of rapid recovery in hog production, the five indicators with the highest explanatory power for recoverability based on the inventory of hogs were industrial structure, slaughter rate, culling rate, economic level, and resource carrying capacity, while the five indicators with the highest explanatory power for recoverability based on the inventory of breeding sows were resource carrying capacity, labor productivity, culling rate, economic level, and industrial structure. The shared indicators were the industrial structure, culling rate, economic level, and resource carrying capacity. The industrial structure reflects the advantages of the hog industry. The higher the industrial structure, i.e., the higher the proportion of the output value of the hog industry in the total agricultural output value is, the more dominant the hog industry is in the region, the better the foundation for its development, and the more robust its recovery and resilience. The culling rate reflects the severity of the impact of African swine fever; hence, the higher the culling rate, the more severe the epidemic and the greater the threat to the resilience of the hog industry. In addition, hogs are grain-fed livestock. Thus, the better the resource base is, the stronger the carrying capacity, the more secure the feed grain for the development of the hog industry, and the stronger its recoverability and resilience. The mechanism of influence of the economic level on the resilience of the hog industry is the same as that noted above.

In 2021, when hog production capacity was reasonably adjusted, the five indicators with the highest explanatory power for recoverability based on the hog inventory were the scale level, slaughter rate, resource carrying, per capita consumption, and economic level, while the five indicators with the highest explanatory power for recoverability based on the breeding sow inventory were the scale level, slaughter rate, per capita consumption, labor productivity, and resource carrying. Contrasting these two sets of indicators, the indicators that apply to both are the scale level, slaughter rate, per capita consumption, and resource carrying capacity. The degree of scale and the slaughter rate reveal the level of scientific and technological support during the reasonable adjustment period of hog production capacity. Scale is the inevitable requirement for the high-quality development of the hog industry, and the slaughter rate reflects the scientific and technological level under normal circumstances. Both had a positive impact on the recoverability and resilience of the hog industry in the adjustment period of hog production capacity. The influence mechanism of per capita consumption and resource carrying capacity is consistent with the above.

Regarding the results of the interaction detection for resistance and recoverability in the hog industry, it can be seen that the effect of any two factors interacting on the resistance and recoverability divergence pattern of the hog industry is greater than the effect of any one factor alone on the spatial divergence pattern of resistance and recoverability, indicating that the relevant effects of the resilience of the hog industry are all integrated effects. In addition, the resistance based on the inventory of hogs underwent a two-factor enhancement of 27 in 2018, and the corresponding number of two-factor enhancements for resistance and recoverability in the period 2019–2021 was 16–17; resistance based on the inventory of breeding sows was a two-factor enhancement of 26 in 2018 and of only 3 in 2019, and the corresponding two-factor enhancements for recoverability in 2020–2021 were all 11; all remaining factor interactions were nonlinear enhancements. For example, the scientific and technological support factor and the epidemic shock factor together affected the resilience of the hog industry. Epidemic shock will inevitably affect the scientific and technological support factor, and when African swine fever occurs, indicators such as the hog slaughter rate will be affected to varying degrees. In 2018–2019, China’s hog slaughter rate was significantly higher than the normal annual level, at 162.04% and 175.32%, respectively.

## 4. Conclusions

In this study, we applied two core variables, namely, the inventory of hogs and that of breeding sows. We measured the industrial resilience and its spatial and temporal characteristics in 31 provinces of China that were faced with the impact of African swine fever, from the perspectives of both resistance and recoverability. We then explored the influence mechanism of the resilience of China’s hog industry using Geodetector, and we obtained the following research conclusions:

First, the periods of 2018–2019 and 2020–2021 represent the resistance and recovery periods of the hog industry under the impact of African swine fever, respectively, with the industry showing poor resilience in the resistance period and good resilience in the recovery period. The increase in the resilience of China’s pork industry was mainly due to the adjustment of the policy for the prevention and control of African swine fever, as well as the implementation of the policy for the recovery of hog breeding and pork production.

Second, there were differences in the resistance and recoverability of the hog industry across different provinces and periods under the impact of African swine fever. In the early stages of the African swine fever outbreak, Tianjin, Shanxi, Guangxi, and Yunnan had strong levels of resistance. In the most severe stage of the African swine fever outbreak, the resistance of the hog industry in all provinces was generally poor. In the period of rapid recovery in hog production, the recoverability of all provinces was very strong. In the reasonable adjustment period of hog production capacity, the recoverability of the 13 provinces, including Henan, Shandong, and other large hog-breeding provinces, was negative based on their inventories of breeding sows, which resulted from active policy regulation.

Third, there were differences in the key factors that influenced the resilience of the hog industry across different provinces and different periods. In the early stages of the African swine fever outbreak, the slaughter rate, economic level, mortality rate, carcass weight, and culling rate had the highest explanatory power for the resistance of the hog industry; in the most severe period of the African swine fever outbreak, the slaughter rate, per capita consumption, and scale level had the highest explanatory power for the resistance of the hog industry. In the period of rapid recovery in hog production, the industrial structure, culling rate, economic level, and resource carrying had the highest explanatory power for the recoverability of the hog industry. Finally, in the reasonable adjustment period of hog production capacity, the scale level, culling rate, per capita consumption, and resource carrying had the highest explanatory power regarding the recoverability of the hog industry.

Based on the above research conclusions, the following countermeasures are suggested: First, the level of resilience of the hog industry should be improved. It is necessary to accelerate the construction of the modern hog industry system, production system, and business system, to promote the modernization of breeding, processing, circulation, sales, and other high-quality development factors of the entire industry chain, and to further enhance the resilience of the industry chain, supply chain, and security level. We need to continuously cultivate new industries, new forms, and new modes, focus on improving the levels of industrial integration and development, further extend the industrial chain, improve the value chain, stabilize the supply chain, and enhance the resilience of the hog industry by internalizing risks. We should strengthen disaster prevention, mitigation, and relief capacity building, support the hog breeding insurance system (particularly regarding the prevention and control of major epidemics), and further consolidate the risk prevention foundation of the hog industry, as well as its risk transfer capacity.

Second, the prevention and control of hog epidemics should be strengthened. The initiative, foresight, and precision of epidemic prevention and control policies should be enhanced, and short-term emergency response, long-term effective response, and market shock regulation and control should be improved. The construction of a hog disease prevention and control system should be promoted by comprehensively covering both the industrial chain and the supply chain, including breeding, slaughtering, processing, circulation, and sales. We need to strengthen the establishment of grassroots animal disease prevention and control teams and enhance the level of technical training. We should optimize biosecurity, culling, vaccination, and other means of prevention and control, and improve epidemic reporting, culling subsidies, and other systems. We should strengthen the integration and demonstration of technologies for the prevention and control of major animal diseases, including African swine fever, and widely promote effective models for epidemic prevention and control. We must accelerate the promotion of innovation in key core technologies for the prevention and control of major animal diseases, and promote scientific research on African swine fever vaccines.

Third, the monitoring and early warning mechanisms should be improved, as should the mechanisms underlying hog epidemic risk identification, market monitoring and early warning, and the regular release of information, reports, and plans. We should concentrate professional talents in the fields of agricultural economics and information technology, rely on big data, cloud computing, and other intelligent technologies, scientifically collect international and domestic information, conduct real-time simulations, and make response plans. Based on one national standard and one set of data, the uniformity and authority of information should be improved. We need to abandon the “internal reference” and other old concepts and old ways, so that the data and information are open and transparent to the whole society, and every producer, consumer, researcher, and policymaker can access the relevant information to avoid market panic and confusion.

## Figures and Tables

**Figure 1 animals-13-02817-f001:**
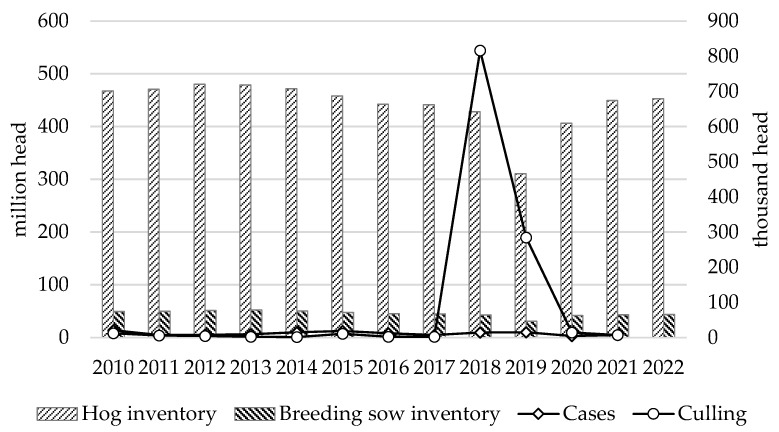
Trends in hog inventory and epidemic factors. Data source: National Bureau of Statistics and the *Official Veterinary Bulletin of the Ministry of Agriculture and Rural Affairs of the People’s Republic of China*.

**Figure 2 animals-13-02817-f002:**
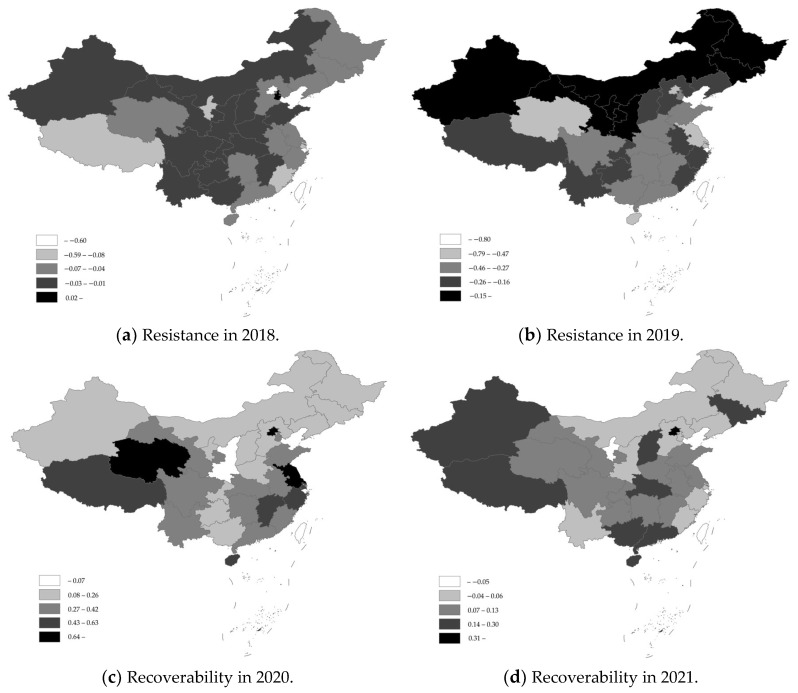
Spatiotemporal characteristics of industry resilience based on the hog inventory.

**Figure 3 animals-13-02817-f003:**
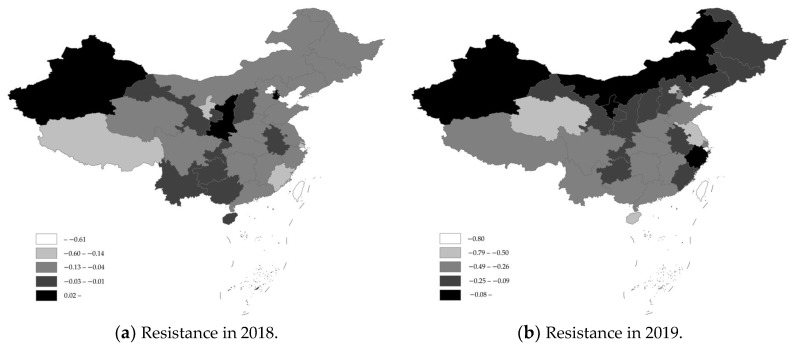
Spatiotemporal characteristics of industry resilience based on the breeding sow inventory.

**Table 1 animals-13-02817-t001:** Geographic detection factors.

Type	Indicator	Unit	Interpretation of Indicators
Development foundation	Economic level	CNY	GDP per capita
	Industrial structure	%	Share of the hog industry’s output value in the total agricultural output value
	Market share	%	Share of the region’s hog inventory in the national inventory
	Per capita consumption	kg	Per capita household consumption of pork
Scientific and technological support	Slaughter rate	%	Ratio of number of hogs slaughtered to the total number of hogs
	Carcass weight	kg/head	Ratio of pork production to the number of hogs slaughtered
	Scale level	%	Share of farms with more than 500 heads in total farms
	Labor productivity	CNY	Ratio of the average gross value added per hog to the number of workers
Basic security	Comparative benefit	—	Ratio of hog prices to corn prices
	Resource carrying	head/ha	Ratio of hog inventory to grain acreage
	Technical service	head	Ratio of hog inventory to the number of employees in the township’s animal husbandry and veterinary stations
Epidemic shock	Cases	head	Number of cases due to African swine fever outbreaks
	Mortality rate	%	Ratio of deaths to total cases due to African swine fever outbreaks
	Culling rate	%	Ratio of culls due to African swine fever outbreaks to total hog inventory

**Table 2 animals-13-02817-t002:** Spatiotemporal characteristics of the resistance and recoverability of the hog industry.

Province	Hog				Breeding Sow			
	Resistance (2018)	Resistance (2019)	Recoverability (2020)	Recoverability (2021)	Resistance (2018)	Resistance (2019)	Recoverability (2020)	Recoverability (2021)
Beijing	−0.5950	−0.7097	1.4397	0.8350	−0.6136	−0.7255	1.1429	0.8333
Tianjin	0.0942	−0.3690	0.3064	0.0545	0.0698	−0.3696	0.3241	−0.0260
Hebei	−0.0700	−0.2210	0.2330	0.0350	−0.0701	−0.1869	0.3225	−0.0160
Shanxi	0.0099	−0.1785	0.2614	0.3000	0.0126	−0.1365	0.3162	0.1139
Inner Mongolia	−0.0164	−0.1362	0.2433	0.0582	−0.0853	−0.0329	0.2571	−0.0226
Liaoning	−0.0350	−0.1640	0.2170	0.0190	−0.0781	−0.1552	0.2562	−0.0152
Jilin	−0.0447	−0.0891	0.1340	0.2653	−0.0855	−0.0946	0.1714	0.2024
Heilongjiang	−0.0563	−0.1330	0.1687	0.0329	−0.0457	−0.1603	0.2203	−0.0270
Shanghai	−0.1327	−0.4741	0.6342	−0.0123	−0.1667	−0.2933	0.7358	−0.3587
Jiangsu	−0.0538	−0.6279	1.3810	0.0784	−0.0578	−0.4996	1.1990	−0.0471
Zhejiang	−0.0475	−0.1732	0.4687	0.0202	−0.0903	−0.0716	0.4453	0.1928
Anhui	−0.0430	−0.1950	0.3000	0.1150	−0.0357	−0.1695	0.4000	0.1088
Fujian	−0.1322	−0.1980	0.4199	0.0293	−0.1667	−0.1906	0.5390	0.0237
Jiangxi	−0.0210	−0.3660	0.5600	0.0722	−0.0525	−0.3210	0.5105	0.1198
Shandong	−0.0180	−0.2710	0.3480	0.0740	−0.0568	−0.3500	0.5499	−0.0701
Henan	−0.0120	−0.2690	0.2260	0.1300	−0.0529	−0.2780	0.3367	−0.0050
Hubei	−0.0220	−0.3585	0.3360	0.1706	−0.0620	−0.3201	0.3582	0.1246
Hunan	−0.0368	−0.2940	0.3840	0.1252	−0.0437	−0.3451	0.4177	0.0469
Guangdong	−0.0509	−0.3411	0.3250	0.1744	−0.0497	−0.3991	0.4099	0.0352
Guangxi	0.0020	−0.3040	0.1430	0.1640	0.0031	−0.3098	0.1679	0.0449
Hainan	−0.0430	−0.5741	0.5262	0.2494	−0.0169	−0.5973	0.6635	0.1425
Chongqing	−0.0205	−0.2104	0.1750	0.0895	−0.0274	−0.2250	0.2392	0.0622
Sichuan	−0.0270	−0.3259	0.3500	0.0980	−0.0630	−0.3199	0.3580	0.0890
Guizhou	−0.0298	−0.2440	0.1646	0.1220	−0.0113	−0.2317	0.2614	0.0377
Yunnan	0.0087	−0.2334	0.3321	0.0639	0.0057	−0.2611	0.2157	0.1103
Tibet	−0.0828	−0.1970	0.6100	0.2373	−0.1519	−0.2687	0.0306	0.3564
Shaanxi	−0.0180	−0.0517	0.0680	0.0418	0.0304	−0.1157	0.0935	0.0391
Gansu	−0.0111	−0.1190	0.2950	0.1014	−0.0112	−0.1570	0.2668	−0.0409
Qinghai	−0.0544	−0.5568	1.0799	0.0712	−0.0889	−0.5244	1.4103	−0.0319
Ningxia	−0.0900	−0.0052	0.2269	−0.0502	−0.1383	0.0864	0.3182	−0.2586
Xinjiang	−0.0201	−0.0862	0.2245	0.1607	0.0633	−0.0174	0.3182	−0.0785

**Table 3 animals-13-02817-t003:** Factor detection results of the resistance and recoverability of the hog industry.

Indicator	Hog				Breeding Sow			
	Resistance (2018)	Resistance (2019)	Recoverability (2020)	Recoverability (2021)	Resistance (2018)	Resistance (2019)	Recoverability (2020)	Recoverability (2021)
Economic level	0.5852	0.3471	0.4294	0.2781	0.5573	0.2012	0.3097	0.1031
Industrial structure	0.0782	0.1208	0.5410	0.1327	0.0943	0.2493	0.2992	0.1079
Market share	0.1868	0.2456	0.3615	0.0926	0.2631	0.2992	0.2543	0.0368
Per capita consumption	0.1035	0.6027	0.3218	0.3112	0.1236	0.5388	0.2130	0.3815
Slaughter rate	0.9143	0.6145	0.4723	0.5590	0.8773	0.4428	0.2553	0.5927
Carcass weight	0.3682	0.1022	0.1749	0.0736	0.4597	0.0478	0.1861	0.1916
Scale level	0.2903	0.3694	0.2574	0.8110	0.3269	0.3312	0.1879	0.7669
Labor productivity	0.0779	0.3073	0.2860	0.2265	0.1430	0.3596	0.3829	0.3338
Comparative benefit	0.0846	0.2528	0.2501	0.2672	0.0875	0.3108	0.1600	0.1848
Resource carrying	0.2238	0.2758	0.4051	0.3662	0.2078	0.3069	0.4692	0.2624
Technical service	0.2637	0.3083	0.3524	0.1621	0.2372	0.3832	0.2419	0.1375
Cases	0.2127	0.3360	0.2037	0.1052	0.2026	0.2583	0.1249	0.0577
Mortality rate	0.3820	0.2555	0.1931	0.1204	0.3770	0.2042	0.1102	0.0205
Culling rate	0.2919	0.2570	0.4323	0.0847	0.3442	0.2326	0.3120	0.0363

**Table 4 animals-13-02817-t004:** Interaction detection results of resistance based on the 2018 hog inventory.

Indicator	Economic Level	Industrial Structure	Market Share	Per Capita Consumption	Slaughter Rate	Carcass Weight	Scale Level	Labor Productivity	Comparative Benefit	Resource Carrying	Technical Service	Cases	Mortality Rate	Culling Rate
Economic level	0.5852													
Industrial structure	0.9802	0.0782												
Market share	0.7055	0.3442	0.1868											
Per capita consumption	0.9991	0.3501	0.2824	0.1035										
Slaughter rate	0.9802	0.9832	0.9924	0.9755	0.9143									
Carcass weight	0.9800	0.9801	0.9271	0.9466	0.9736	0.3682								
Scale level	0.9816	0.3462	0.3475	0.3444	0.9824	0.9899	0.2903							
Labor productivity	0.9858	0.2378	0.2996	0.3474	0.9926	0.5296	0.3377	0.0779						
Comparative benefit	0.9852	0.3486	0.2950	0.3497	0.9684	0.9907	0.3430	0.2181	0.0846					
Resource carrying	0.9681	0.4041	0.9960	0.9459	0.9746	0.6889	0.9840	0.9918	0.5643	0.2238				
Technical service	0.9948	0.9871	0.4900	0.5432	0.9709	0.9659	0.9898	0.6305	0.6205	0.9616	0.2637			
Cases	0.9289	0.9800	0.9964	0.9992	0.9929	0.5427	0.9980	0.5284	0.9803	0.9971	0.5424	0.2127		
Mortality rate	0.9993	0.9742	0.6186	0.5666	0.9760	0.9809	0.9920	0.9821	0.9836	0.9806	0.9720	0.9781	0.3820	
Culling rate	0.9968	0.9957	0.9987	0.5722	0.9819	0.7074	0.9983	0.9998	0.4731	0.7071	0.9682	0.9700	0.9979	0.2919

Note: two-factor enhancement values are underlined, while other values are from the nonlinear enhancement, the same as Table 5, Table 6, Table 7, Table 8, Table 9, Table 10 and Table 11.

**Table 5 animals-13-02817-t005:** Interaction detection results of resistance based on the 2019 hog inventory.

Indicator	Economic Level	Industrial Structure	Market Share	Per Capita Consumption	Slaughter Rate	Carcass Weight	Scale Level	Labor Productivity	Comparative Benefit	Resource Carrying	Technical Service	Cases	Mortality Rate	Culling Rate
Economic level	0.3471													
Industrial structure	0.8503	0.1208												
Market share	0.8542	0.6745	0.2456											
Per capita consumption	0.9610	0.9765	0.9552	0.6027										
Slaughter rate	0.9366	0.9687	0.9612	0.8183	0.6145									
Carcass weight	0.8073	0.8968	0.9917	0.9019	0.9482	0.1022								
Scale level	0.7635	0.8041	0.9995	0.9535	0.9922	0.8350	0.3694							
Labor productivity	0.8163	0.7001	0.8338	0.9724	0.9487	0.9512	0.7941	0.3073						
Comparative benefit	0.9491	0.9017	0.9009	0.9465	0.9846	0.8537	0.7758	0.8685	0.2528					
Resource carrying	0.9804	0.8120	0.8075	0.7923	0.8318	0.8958	0.9946	0.9205	0.8821	0.2758				
Technical service	0.8629	0.6026	0.8374	0.9613	0.9448	0.8211	0.7291	0.7987	0.8462	0.9736	0.3083			
Cases	0.9198	0.9655	0.8692	0.9373	0.9236	0.5746	0.9506	0.8072	0.8640	0.8538	0.9692	0.3360		
Mortality rate	0.7950	0.6482	0.9939	0.8608	0.8972	0.8553	0.8228	0.6949	0.8294	0.8886	0.9483	0.6319	0.2555	
Culling rate	0.7848	0.7632	0.9478	0.9355	0.8238	0.7633	0.8214	0.7298	0.7716	0.9423	0.9725	0.6897	0.5375	0.2570

**Table 6 animals-13-02817-t006:** Interaction detection results of recoverability based on the 2020 hog inventory.

Indicator	Economic Level	Industrial Structure	Market Share	Per Capita Consumption	Slaughter Rate	Carcass Weight	Scale Level	Labor Productivity	Comparative Benefit	Resource Carrying	Technical Service	Cases	Mortality Rate	Culling Rate
Economic level	0.4294													
Industrial structure	0.8442	0.5410												
Market share	0.8553	0.8768	0.3615											
Per capita consumption	0.9839	0.9871	0.9924	0.3218										
Slaughter rate	0.9601	0.8817	0.6651	0.9981	0.4723									
Carcass weight	0.9132	0.9824	0.7597	0.5763	0.9690	0.1749								
Scale level	0.9151	0.9465	0.9484	0.9598	0.9540	0.8814	0.2574							
Labor productivity	0.8982	0.9994	0.9998	0.8329	0.9955	0.6587	0.8083	0.2860						
Comparative benefit	0.9755	0.8917	0.8983	0.6917	0.8345	0.8664	0.6607	0.9762	0.2501					
Resource carrying	0.9558	0.8272	0.9828	0.8476	0.9895	0.8444	0.9738	0.7536	0.9092	0.4051				
Technical service	0.9464	0.8713	0.6552	0.8677	0.6803	0.6976	0.7886	0.9946	0.7875	0.9670	0.3524			
Cases	0.7452	0.7725	0.7384	0.6957	0.8885	0.5009	0.9262	0.8400	0.7873	0.9833	0.5157	0.2037		
Mortality rate	0.9999	0.8796	0.7669	0.9824	0.8413	0.8160	0.5150	0.7551	0.8474	0.9993	0.7286	0.7830	0.1931	
Culling rate	0.8504	0.8823	0.8504	0.6835	0.8911	0.6396	0.6592	0.6717	0.8272	0.9773	0.8905	0.6857	0.6993	0.4323

**Table 7 animals-13-02817-t007:** Interaction detection results of recoverability based on the 2021 hog inventory.

Indicator	Economic Level	Industrial Structure	Market Share	Per Capita Consumption	Slaughter Rate	Carcass Weight	Scale Level	Labor Productivity	Comparative Benefit	Resource Carrying	Technical Service	Cases	Mortality Rate	Culling Rate
Economic level	0.2781													
Industrial structure	0.9189	0.1327												
Market share	0.4921	0.3213	0.0926											
Per capita consumption	0.8924	0.9019	0.9285	0.3112										
Slaughter rate	0.9764	0.9345	0.6849	0.9480	0.5590									
Carcass weight	0.9938	0.9085	0.9351	0.5960	0.9777	0.0736								
Scale level	0.8972	0.9245	0.9647	0.9477	0.9033	0.9919	0.8110							
Labor productivity	0.9624	0.9949	0.7545	0.9849	0.7154	0.5853	0.9265	0.2265						
Comparative benefit	0.9302	0.9443	0.8962	0.9772	0.9440	0.8873	0.9795	0.9924	0.2672					
Resource carrying	0.9947	0.8981	0.9552	0.8841	0.9846	0.8789	0.9588	0.9680	0.6356	0.3662				
Technical service	0.9680	0.3570	0.3945	0.9555	0.9560	0.9893	0.9440	0.9930	1.0000	0.9865	0.1621			
Cases	0.4991	0.4157	0.3126	0.9599	0.6538	0.4626	0.9427	0.4628	0.9207	0.9340	0.4452	0.1052		
Mortality rate	0.4724	0.4545	0.3005	0.9671	0.9673	0.5552	0.9280	0.5847	0.9946	0.9946	0.4696	0.2372	0.1204	
Culling rate	0.4874	0.4133	0.3180	0.5978	0.7545	0.3046	0.9793	0.5333	0.9097	0.9325	0.4360	0.2323	0.2814	0.0847

**Table 8 animals-13-02817-t008:** Interaction detection results of resistance based on the 2018 breeding sow inventory.

Indicator	Economic Level	Industrial Structure	Market Share	Per Capita Consumption	Slaughter Rate	Carcass Weight	Scale Level	Labor Productivity	Comparative Benefit	Resource Carrying	Technical Service	Cases	Mortality Rate	Culling Rate
Economic level	0.5573													
Industrial structure	0.9366	0.0943												
Market share	0.7490	0.4376	0.2631											
Per capita consumption	0.9997	0.4432	0.3925	0.1236										
Slaughter rate	0.9307	0.9453	0.9968	0.9891	0.8773									
Carcass weight	0.9553	0.9759	0.9240	0.9744	0.9658	0.4597								
Scale level	0.9369	0.4421	0.4383	0.4235	0.9819	0.9872	0.3269							
Labor productivity	0.9682	0.3445	0.4154	0.4235	0.9882	0.6452	0.3965	0.1430						
Comparative benefit	0.9665	0.4445	0.4095	0.4452	0.9886	0.9846	0.4136	0.3192	0.0875					
Resource carrying	0.9563	0.4421	0.9964	0.9019	0.9693	0.7420	0.9690	0.9734	0.5789	0.2078				
Technical service	0.9825	0.9300	0.6013	0.6207	0.9792	0.9756	0.9911	0.7334	0.7175	0.9440	0.2372			
Cases	0.9234	0.9713	0.9943	0.9737	0.9917	0.6553	0.9900	0.6608	0.9566	0.9663	0.4300	0.2026		
Mortality rate	0.9945	0.9667	0.7011	0.5603	0.9783	0.9825	0.9763	0.9684	0.9694	0.9691	0.9568	0.9628	0.3770	
Culling rate	0.9576	0.9888	0.9828	0.6567	0.9950	0.7497	0.9921	0.9994	0.5040	0.7607	0.9805	0.9397	0.9903	0.3442

**Table 9 animals-13-02817-t009:** Interaction detection results of resistance based on the 2019 breeding sow inventory.

Indicator	Economic Level	Industrial Structure	Market Share	Per Capita Consumption	Slaughter Rate	Carcass Weight	Scale Level	Labor Productivity	Comparative Benefit	Resource Carrying	Technical Service	Cases	Mortality Rate	Culling Rate
Economic level	0.2012													
Industrial structure	0.8782	0.2493												
Market share	0.8121	0.6779	0.2992											
Per capita consumption	0.9210	0.9950	0.9746	0.5388										
Slaughter rate	0.9070	0.9606	0.9424	0.7702	0.4428									
Carcass weight	0.7520	0.8966	0.9841	0.8824	0.9152	0.0478								
Scale level	0.6932	0.8015	0.9999	0.8927	0.9648	0.9014	0.3312							
Labor productivity	0.7937	0.7634	0.8419	0.9784	0.9196	0.9417	0.8351	0.3596						
Comparative benefit	0.9307	0.8702	0.8794	0.9329	0.9604	0.8848	0.7667	0.8814	0.3108					
Resource carrying	0.9804	0.7734	0.7805	0.7829	0.8190	0.9096	0.9889	0.9488	0.8894	0.3069				
Technical service	0.8852	0.6883	0.8675	0.9842	0.9049	0.8445	0.8155	0.8403	0.8884	0.9356	0.3832			
Cases	0.8610	0.9450	0.9207	0.9205	0.9409	0.4364	0.8560	0.7935	0.8351	0.8929	0.9897	0.2583		
Mortality rate	0.7492	0.6003	0.9969	0.8896	0.8818	0.7904	0.7033	0.6977	0.7521	0.9226	0.9467	0.5767	0.2042	
Culling rate	0.7258	0.7061	0.9453	0.8718	0.7927	0.6944	0.6648	0.7283	0.8326	0.9471	0.9662	0.6909	0.4879	0.2326

**Table 10 animals-13-02817-t010:** Interaction detection results of recoverability based on the 2020 breeding sow inventory.

Indicator	Economic Level	Industrial Structure	Market Share	Per Capita Consumption	Slaughter Rate	Carcass Weight	Scale Level	Labor Productivity	Comparative Benefit	Resource Carrying	Technical Service	Cases	Mortality Rate	Culling Rate
Economic level	0.3097													
Industrial structure	0.5905	0.2992												
Market share	0.6519	0.6376	0.2543											
Per capita consumption	0.9835	0.9738	0.9093	0.2130										
Slaughter rate	0.6811	0.6170	0.4633	0.9946	0.2553									
Carcass weight	0.7490	0.9793	0.8482	0.5788	0.9652	0.1861								
Scale level	0.6343	0.6752	0.6755	0.9464	0.6623	0.7147	0.1879							
Labor productivity	0.7989	0.9917	0.9997	0.8701	0.9975	0.8215	0.8806	0.3829						
Comparative benefit	0.9657	0.7928	0.7982	0.6193	0.7123	0.9051	0.6397	0.9543	0.1600					
Resource carrying	0.9699	0.8585	0.9895	0.8756	0.9826	0.9009	0.9780	0.8442	0.7861	0.4692				
Technical service	0.6768	0.6131	0.4479	0.7485	0.4772	0.7729	0.5828	0.9993	0.6017	0.9379	0.2419			
Cases	0.5183	0.5550	0.5508	0.6273	0.6326	0.6432	0.6379	0.8849	0.5915	0.9840	0.3900	0.1249		
Mortality rate	0.9986	0.7365	0.5808	0.9675	0.7035	0.9207	0.5556	0.8834	0.7311	1.0000	0.6963	0.5919	0.1102	
Culling rate	0.5978	0.6318	0.5764	0.6267	0.6216	0.5496	0.4501	0.7089	0.7626	0.9583	0.6255	0.4915	0.7165	0.3120

**Table 11 animals-13-02817-t011:** Interaction detection results of recoverability based on the 2021 breeding sow inventory.

Indicator	Economic Level	Industrial Structure	Market Share	Per Capita Consumption	Slaughter Rate	Carcass Weight	Scale Level	Labor Productivity	Comparative Benefit	Resource Carrying	Technical Service	Cases	Mortality Rate	Culling Rate
Economic level	0.1031													
Industrial structure	0.9572	0.1079												
Market share	0.3173	0.4077	0.0368											
Per capita consumption	0.9015	0.8801	0.9889	0.3815										
Slaughter rate	0.9492	0.8861	0.8414	0.8835	0.5927									
Carcass weight	0.9690	0.9867	0.8270	0.6734	0.9748	0.1916								
Scale level	0.8572	0.9575	0.9208	0.9401	0.9560	0.9813	0.7669							
Labor productivity	0.9699	0.9988	0.8819	0.9508	0.8705	0.6763	0.9299	0.3338						
Comparative benefit	0.9052	0.9664	0.7257	0.9515	0.8528	0.7852	0.9708	0.9900	0.1848					
Resource carrying	0.9982	0.9781	0.9976	0.9179	0.8335	0.8329	0.9557	0.8954	0.5841	0.2624				
Technical service	0.9220	0.3384	0.3863	0.9764	0.9827	0.9141	0.8859	0.9813	1.0000	0.9741	0.1375			
Cases	0.3648	0.4755	0.1946	0.9900	0.7962	0.4750	0.9667	0.6419	0.7386	0.9817	0.4786	0.0577		
Mortality rate	0.3659	0.3978	0.2452	0.9829	0.9339	0.6415	0.9901	0.6783	0.9940	0.9962	0.4073	0.1913	0.0205	
Culling rate	0.3624	0.4291	0.1918	0.7476	0.8490	0.3919	0.9542	0.6034	0.7175	0.9017	0.4626	0.1602	0.2400	0.0363

## Data Availability

Data are available upon request from the authors.

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
