# Peer review of "African Swine Fever Shock: China’s Hog Industry’s Resilience and Its Influencing Factors"

_animals, 2023, doi:10.3390/ani13182817_

Round 1
Reviewer 1 Report
The team of two authors submitted a manuscript that is interesting, and it is linked to the objectives of the journal, however, there are some issues that have to be reconsidered.
The objective of the manuscript is to provide original insight into measuring the industrial resilience of 31 Chinese provinces which faced the African swine fever shock and its spatial and temporal differentiation characteristics from 2018-2021 in terms of the two dimensions of resistance and recoverability.
The subject area is rather interesting, and, possibly, not enough approached by other scholars, so there is potential room for this manuscript to bring new information, once it reaches the expected level of quality.
The Abstract has to be reconsidered. The general information provided must be condensed as this part is too large. The Abstract fails in presenting the main findings on commensurable terms.
For better visibility on databases, the authors are asked not to repeat among keywords the words/concepts included in the title of the article. Entering different words in the title and in the keywords can improve the search for the paper in metasearch engines and internet databases.
In the introduction, some data seems not to have references (see the first two paragraphs of the Instruction). The objective of the study is clearly enough pointed out, explained and defined. The part of the Literature Review constructed a reliable scientific gap to be covered by the manuscript.
The methodology part is offering the necessary information.
The Results part is the longest one in the manuscript and it pointed out relevant information.
- Figures 1, 2 and 4. It is advisable to use more distinct patterns among the variables (many different colors?)
- Tables 2-11 are long and difficult to follow. Please point out the main findings in the tables
The discussion part is consistent and brings important information for scholars.
The conclusion part is a bit too long (I recommend reconsidering it by moving the information related to the discussion of the results).
Reviewer 2 Report
The article, African Swine Fever Shock: China’s Hog Industry Resilience and its Influencing Factors, by Shi et al., is well-written and covers data from across 31 provinces in China. It highlights the effects of government policies, direct disease impact on the hog industry, the journey to recovery, and future recommendations. While there have been some published articles regarding the ASF situation in China, the authors added new information and used data from credible sources to make this article interesting.
Minor comments:
Line 58: Define CPI
Figure 1: Visualizing the “Cases” data is difficult because the plot is so close to the baseline, especially from 2018 to 2021. The figure can be improved by using two segments in the y-axis, allowing the “0-100 million heard” scale to use about 30% of the y-axis; this way, the data would be more visible and not appear to be close to zero.
Reviewer 3 Report
The MS measured the industrial resilience 31 Chinese provinces faced the African swine fever shock and its spatial and temporal differentiation characteristics from 2018-2021 in terms of the two dimensions of resistance and recoverability. It could help in improving prevent and mitigate epidemic risks, and to promote high quality development of the hog industry, and meeting the nutritional needs of residents.
The MS is well organized and prepared.
There are too many tables displayed in the MS. The authors should decide what contents needed to be showed to the readers.
NA
Reviewer 4 Report
In this work, titled "African Swine Fever Shock: China's Hog Industry Resilience and its Influencing Factors," the authors measured the industrial resilience of 31 Chinese provinces facing the African swine fever shock and its spatial and temporal differentiation characteristics from 2018-2021 in terms of the two dimensions of resistance and recoverability. Through the use of Geodetector, the study explored the key factors influencing the resilience of China's hog industry. The work provides an interesting methodology to assess the resilience of China's hog industry and explore the relevant influencing factors, which can guide the implementation of future countermeasures.
1. The introduction could be revised in terms of discussing the existing literature. The authors can add further literature review on the resilience of the hog industry, or the livestock industry, or the agricultural sector and its influencing, around the research topic.
2. In Results and Discussion, the results of the interaction detection of resistance and recoverability should be further discussed and analyzed in depth, the reasoning behind the results could be described beyond stating the statistics.
3. In Conclusions, the countermeasures could be described in more detail. In particular, when discussing policy recommendations to strengthen the prevention and control of pig epidemics and to improve the monitoring and early warning mechanism, more specific and actionable recommendations should be made.
